# A Retrospective Study about the Differences in Cardiometabolic Risk Indicators and Level of Physical Activity in Bariatric Surgery Patients from Private vs. Public Units

**DOI:** 10.3390/ijerph16234751

**Published:** 2019-11-27

**Authors:** Rebeca Rocha de Almeida, Márcia Ferreira Cândido de Souza, Dihogo Gama de Matos, Larissa Monteiro Costa Pereira, Victor Batista Oliveira, Joselina Luzia Menezes Oliveira, José Augusto Soares Barreto-Filho, Marcos Antonio Almeida-Santos, Raphael Fabrício de Souza, Aristela de Freitas Zanona, Victor Machado Reis, Felipe J. Aidar, Antônio Carlos Sobral Sousa

**Affiliations:** 1Post Graduate Program in Health Sciences, Federal University of Sergipe—UFS, Aracaju, Sergipe 49060-108, Brazil; rebeca_nut@hotmail.com (R.R.d.A.); nutrimarciacandido@gmail.com (M.F.C.d.S.); larissa_monteiroo@hotmail.com (L.M.C.P.); vbo.nutri@gmail.com (V.B.O.); joselinamenezes@gmail.com (J.L.M.O.); acssousa@terra.com.br (A.C.S.S.); 2Estácio Sergipe University Center, Aracaju, Sergipe 49020-490, Brazil; 3Group of Studies and Research in Performance, Sport, Health and Paralympic Sports—GEPEPS, Federal University of Sergipe—UFS, São Cristovão, Sergipe 49100-000, Brazil; dihogogmc@hotmail.com (D.G.d.M.); raphaelctba20@hotmail.com (R.F.d.S.); 4Institute of Parasitology, McGill University, Montreal, QC H3A 0E6, Canada; 5Department of Medicine, Federal University of Sergipe, Aracaju, Sergipe 49060-108, Brazil; 6Cardiovascular System Unit Federal University of Sergipe, Aracaju, Sergipe 49060-108, Brazil; 7Clinic and Hospital São Lucas—Rede D’Or São Luiz, Aracaju, Sergipe 49015-400, Brazil; 8Post Graduate Program in Health and Environment, Tiradentes University, Aracaju, Sergipe 49032-490, Brazil; marcosalmeida2010@yahoo.com.br; 9Department of Occupational Therapy, Federal University of Sergipe—UFS, Lagarto, Sergipe 49170-000, Brazil; arisz_to@yahoo.com.br; 10Research Center in Sports Sciences, Health Sciences and Human Development, CIDESD, 5000-801 Vila Real, Portugal; victormachadoreis@gmail.com; 11Department of Physical Education, Federal University of Sergipe—UFS, São Cristovão, Sergipe 49060-108, Brazil; 12Graduate Program in Physiological Sciences, Federal University of Sergipe—UFS, São Cristovão, Sergipe 49100-000, Brazil

**Keywords:** obesity, bariatric surgery, health, cardiometabolic risk

## Abstract

*Background*: Obesity is a pathology with a growing incidence in developing countries. *Objective*: To evaluate the evolution of cardiometabolic, anthropometrics, and physical activity parameters in individuals undergoing bariatric surgery (BS) in the public healthcare system (PUS) and private healthcare system (PHS). *Methods*: A longitudinal, observational, and retrospective study was conducted with 111 bariatric patients on two different health systems, with 60 patients from the PUS and 51 from the PHS. Cardiometabolic risk (CR) was analyzed by the assessment of obesity-related comorbidities (AORC) on admission and 3, 6, and 12 months after BS, and the International Physical Activity Questionnaire (IPAQ) was surveyed before and 12 months after BS. In addition, cardiometabolic risk was also assessed by biochemical (fasting glucose and complete lipidogram) and anthropometric (weight, weight loss, waist circumference, and waist-to-height ratio) parameters. *Results*: On admission, the parameters of severe obesity, systemic arterial hypertension (SAH), Diabetes mellitus (DM), and waiting time to BS were higher in the PUS. Additionally, in the PUS, AORC was reduced only in the SAH parameter. However, in the post-surgery moment, AORC reduced, and there was no difference between the two groups after BS. Regarding physical activity, the IPAQ showed a higher level of activity in the PHS before and one year after BS. *Conclusion*: At the PUS, BS is performed in patients with a higher degree of comorbidities, but BS improved the reduction of the CR at a similar level to those observed in the PHS.

## 1. Background

The prevalence of obesity has considerably increased in the last three decades, affecting 13% of the adult population, both in developed and developing nations. It is a major risk factor for an increasing group of chronic pathologies, including cardiovascular diseases, diabetes mellitus, kidney disease, different kinds of cancer, and several musculoskeletal disorders [1].

In Brazil, this condition affects 18.9% of the population, almost one in five Brazilians [2]. Due to the recommendation of the Brazil Ministry of Health [3], surgical interventions for morbid obesity are standardized and performed by the public healthcare system (PUS). Currently, according to the International Federation of Obesity Surgery and Metabolic Disorders, Brazil performed in 2014 a total of 86,840 bariatric surgeries (BSs), being the second place in numbers of BS, only behind North America (USA/CANADA). Despite the growth in the numbers of BS performed in the PUS, the Private Healthcare System (PHS) performed even more BSs, as shown by the National Supplementary Health Agency (ANS) of Brazil, which only in 2017 performed 48,299 BSs. The PUS covers almost 75% of the general population, and the PHS has a mere 50 million beneficiaries [3,4]. However, Brazil’s health system surpasses the boundaries of the PUS, being formed by a complex network of service providers and owners, forming a public-private arrangement [5].

BS appears to be an effective tool for the treatment of obesity and for obesity-related comorbidities, including SAH, DM, dyslipidemia, and cardiovascular events, as well as a means to reduce mortality [6,7,8]. Conservative treatment, diet, and exercise have been established to induce significant weight loss and increase aerobic fitness along with a cardiometabolic benefit [7]. However, the low long-term adherence of individuals to this approach may lead to the recovery of body weight in 95% of the patients [4,7]. On the other hand, the surgical procedure does not minimize the risks associated with a sedentary lifestyle [9], and a lifestyle change after surgery is essential for optimal outcomes [10,11].

At the moment, one of the most important aspects for defining success after bariatric surgery is the long-term effect on weight reduction in body fat. Therefore, favorable changes in body composition (decreased adiposity and maintenance of fat-free mass) are desirable since they are associated with reduced cardiovascular risk, especially coronary atherosclerosis. In addition to body weight, indicators, such as body mass index (BMI), excessive weight loss (EWL), waist circumference, and waist-to-height ratio, can also predict cardiovascular risk and are often used to monitor bariatric surgery [12].

Studies show that Brazilians depending on the PUS (71%) have a profile characterized by physical inactivity and lower education and income. Therefore, scientific evidence shows that weight loss and a slight increase in activity level can improve the physical condition, glucose metabolism, corporal composition, and quality of life [13].

The goal of the present research was to evaluate changes in anthropometrics, cardiometabolic, and physical activity patterns of bariatric surgery patients on the PUS and PHS, pre-surgery, and up to 12-months post-surgery.

## 2. Methods

This research followed the components of the Protocol Strengthening the Reporting of Observational Studies in Epidemiology (STROBE) for observational studies, and the procedures followed the schematic presented in Figure 1.

## 3. Sample

The sample consisted of 111 patients who underwent BS between 2007 and 2016. The calculation of the minimum sample size was made by GPower^®^ software [14]. According to the paired and two-tailed tests, adopting a significance level of 5% with 80% power, a representative sample can be obtained with 60 patients from the PUS and 51 from the PHS [14].

The eligibility criteria of the patients included age between 18 and 65 years at the time of surgery, classified as eligible for surgery in accordance to the Resolution of the Federal Council of Medicine (FCM 2131/2015) [15], and presenting the following comorbidities associated with obesity: hypertension, dyslipidemia, and/or diabetes mellitus, with at least one year of post-surgery and complete follow-up data in their medical records. Patients who abandoned medical and nutritional monitoring within one year of surgery and/or had incomplete data needed for the study were excluded, along with women who became pregnant during the first year after the surgery.

Research ethical standards were followed according to resolution No.510, of 04/07/2016 of the National Council of Health, normative of research that involves the use of data, and agrees with the ethical principles contained in the Declaration of Helsinki (2013). All participants signed the consent term. This study was approved by the Research Ethics Committee (CEP) of the Federal University of Sergipe under the number 1276237.

## 4. Data Collection

### 4.1. Anthropometrics

The data collection was performed retrospectively utilizing records in charts used in the nutritional assistance of patients undergoing BS. The anthropometric data were related to height, weight at admission, pre-surgery weight, post-surgery weight at 12 months, weight loss percentage, waist circumference at admission, waist circumference in post-surgery at 12 months, and waist-to-height ratio.

### 4.2. Comorbidities Related to Obesity

The clinical progression of the cardiometabolic risk factors (CRF) was quantified as proposed in previous studies by Ali et al. [10] and Silva-Neto et al. [16], calculating scores for the Evaluation of Comorbidities Related to Obesity (AORC). This was based on a points system that assigned scores from 0 to 5, according to severity, for the components of the CRF: Diabetes mellitus, dyslipidemia, and hypertension (Table 1).

According to AORC, a score ≥ 3 indicates that the patient needs medical treatment or has complications related to the disease. We adopted cutoff points for ≤2 and ≥3 for the absence and presence of obesity-associated comorbidities, respectively, according to the study by Farinholt et al. [17]. The scores were assigned at the time of admission, pre-surgery, and in the post-surgery follow-ups (approximately 3, 6, and 12 months).

### 4.3. Biochemical Analysis

The biochemical data collected were related to serum and/or plasma levels of triglycerides, total cholesterol, high density lipoproteins (HDL-cholesterol), low density lipoproteins (LDL-cholesterol), and fasting glucose. The recording of these parameters was related to three moments: preoperative, and 6 and 12 months postoperative.

### 4.4. International Physical Activity Questionnaire (IPAQ)

The questionnaire contained questions related to physical activities performed in the last week prior to the application of the questionnaire. The responses of the individuals were analyzed, and the individuals were classified according to the questionnaire, which divides and conceptualizes the categories as follows:

***Sedentary***: individuals who do not perform any physical activity for at least 10 min continuously during the week;

***Insufficiently Active***: individuals who engage in physical activity for at least 10 min continuously per week, but insufficiently to be classified as active. For inclusion in this category, the duration and frequency of the different types of activities (moderate + vigorous + walks) are added. This category is divided into the two following groups:

***Insufficiently Active A***: individuals who perform 10 min of continuous physical activity, adhering to at least one of the following criteria: A frequency of 5 days/week or duration of 150 min/week;

***Insufficiently Active B***: individuals who do not meet any of the specifications of the individuals in the Insufficiently Active A category. 

***Active***: individuals who meet the following recommendations: (a) vigorous physical activity ≥ 3 days/week and >20 min/session; (b) moderate exercise or walking ≥ 5 days/week and >30 min/session; (c) any activity added to >5 days/week and >150 min/week; 

***Very Active***: individuals who meet the following recommendations: (a) vigorous ≥ 5 days/week and >30 min/session; (b) vigorous ≥ 3 days/week and >20 min/moderate + and/or a walk 3 to 5 days/week for a >30 min/session.

***Vigorous***: ≥5 days/week and >30 min/session; (b) vigorous ≥ 3 days/week and >20 min/moderate + and/or a walk 3 to 5 days/week for a >30 min/session.

### 4.5. Statistics

The central tendency measures and mean ± standard deviation (X ± SD) were used. Analyses were performed according to the nature of the variables, and those that met the normality assumption were expressed as mean and standard deviation while the others were demonstrated in median (1st and 3rd quartiles). Categorical variables were summarized as simple frequency and percentage. 

For the comparisons between the PUS and PHS groups, following a normality assumption verification (Kolmogorov–Smirnov), Student’s t-test was used for independent data, and for those that did not meet this assumption, the Mann–Whitney test was used.

The analysis of the association between the healthcare system groups and the qualitative variables (categorical) was performed using the Chi-square or Fisher’s exact test. To evaluate the AORC score over time, the Cochran Q test followed by the McNemar test was used to analyze the changes of patients in hospital groups using the RCM categories (≤2 and ≥3), comparing the various time points (admission, pre, and post-surgery at 3, 6, and 12 months). The reliability study of the AORC score instrument was measured by the Cronbach α index, which was 0.895 with a variation of 0.869–0.882 based on the standardized items, followed by the ANOVA with Cochran’s Q test with its 95% confidence interval.

To compare levels of physical activity and Body Mass Index two-way ANOVA (System x Moment) was conducted with Tukey post-hoc, after normality (Kolmogorov–Smirnov) and sphericity (Mauchly) assumptions were confirmed. The effect size was calculated by the partial Eta^2^ (hp^2^), considering ≤0.05 (small effect), between 0.05 and 0.25 (medium effect), among 0.25 and 0.50 (high effect), and finally >0.5 (very high effect) [18]. The level of significance was set at 5%. Statistical analyses were performed using the Statistical Package for the Social Science, SPSS, Version 22.0 for Windows.

## 5. Results

The sample of patients undergoing BS in the PUS and PHS of the study consisted of 111 individuals with a mean age of 39.6 ± 10.8 years, ranged from 19 to 66 years; the mean age in the PUS network was significantly higher (*p* = 0.048) than that of the PHS. Among the comorbidities, patients in the PUS network had a higher prevalence of SAH (*p* = 0.008) and diabetes mellitus (*p* = 0.018) when compared with the PHS. 

The characterization of the sample is shown in Table 2. Most of the individuals in the study were women (72.1%), and the distribution was similar between the groups. The time elapsed between admission and surgery was significantly higher (*p* < 0.001) for the PUS patients (Table 2).

Regarding the nutritional status in the period of admission to services, the majority of PUS patients had a classification of severe obesity (grade III obesity) when compared with those in the PHS (Table 3). 

The progression of PUS patients after undergoing BS appears in Figure 2. In the admission and pre-surgery period, there were no significant changes in the AORC score ≥ 3 of the factors associated with CRF; changes in dyslipidemia (*p* < 0.0001) and SAH (*p* < 0.0001) were observed in the pre- and post-surgery period at three months, a significant reduction in the prevalence (or frequency) of diabetes mellitus. In subsequent post-surgery periods at 6 and 12 months, there was no significant further progression concerning the comorbidities studied.

The progression of PHS patients undergoing BS is shown in Figure 2. In the admission and pre-surgery period, there was no significant change in the AORC score ≥ 3 in relation to diabetes mellitus and dyslipidemia; in contrast, there were changes in CRF at that time (*p* = 0.004). 

In the period between the pre and post-surgery at three months, there was a decrease in CRF (*p* = 0.001), with no change in diabetes mellitus and dyslipidemia. In the later periods (3, 6, and 12 months), there was no significant change in their comorbidities.

The comparison between the PUS and PHS groups concerning the anthropometric changes are described in Table 4. A significant difference was observed pre- and post-surgery in the nutritional status of the patients (*p* < 0.0001), in which more patients in the PUS network were classified as having more severe degrees of obesity (grade II and III obesity). The patients followed up by PUS had a higher mean pre-surgery weight (*p* < 0.001), post-surgery weight (after 12 months of surgery) (*p* < 0.001), pre-surgery BMI (*p* < 0.001), post-surgery BMI (*p* < 0.001), and initial and final waist circumference (*p* < 0.001). Only the weight loss percentage of PHS (*p* = 0.002) was significantly higher than that of the PUS network (Table 4).

Regarding the biochemical variables, according to the periods, the comparison between the PUS and PHS groups is described in Table 5 and Table 6. LDL-c, cholesterol, triglycerides, and fasting glucose levels decreased and HDL-c increased, and in HDL, LDL, and cholesterol, there was a significant reduction between groups (*p* = 0.014; *p* = 0.046; *p* < 0.0001), however the effect size was considered large, medium, and insignificant (*η*^2^ = 0.080; *η*^2^ = 0.058; *η*^2^ = 0.063), respectively. Regarding the time between groups, there was a significant reduction in LDL-c, cholesterol, triglycerides, and fasting glucose (*p* < 0.0001), but the effect size was considered small. (*η*^2^ = 0.197–0.455).

Regarding the level of physical activity (IPAQ) and body mass index (BMI) between the individuals from PUS and PHS submitted to the BS, they were compared in the moment of admission and one year after BS (Figure 3 and Figure 4). 

The IPAQ analysis verified that there were no differences in the patients on the admission in the PUS before (119.57 ± 32.55 min, IC 95% 107.39–139.75) and PUS after (195.43 ± 50.02 min, IC 95% 178.95–212.61). There were differences between PUS before and PHS after (248.86 ± 147.21 min, IC 95% 168.29–299.42).

Moreover, according with Table 6. there were also differences between PHS before (114.57 ± 155.48 min, IC 95% 61.16–167.98), in comparison with the PHS and PUS after, with *p* value of <0.001, F (1.36) = 39.472, hp^2^ = 0.523 (very high effect). There was no difference between PUS and PHS before with a *p* value of 0.862, F (1.36) = 0.240. hp^2^ = 0.038 (small effect). There were also no differences between the PUS and PHS after, as shown by *p* = 0.220, F (1.36) = 0.240, hp^2^ = 0.038 (small effect).

Still regarding Figure 4, there were no differences in the patients BMI on admission in the PUS before (51.94 ± 9.54 Kg/m^2^, IC 95% 49.26–54.62) and PUS after (34.02 ± 7.41 Kg/m^2^, IC 95% 31.93–36.10), or the BMI PHS after (28.29 ± 4.35 Kg/m^2^, IC 95% 27.07–29.52), with *p* < 0.001, F (1.50) = 35.34, *ηp*^2^ = 0.414 (high effect). There were differences too on the PHS before (40.99 ± 5.38 Kg/m^2^, IC 95% 39.7–42.50), in relation to PHS and PUS after, as shown by *p* < 0.001, F (1.50) = 35.34, *ηp*^2^ = 0.414 (high effect). There was no difference among PUS and PHS before as verified by *p* = 0.999, F (1.50) = 0.040, *ηp*^2^ = 0.067 (medium effect). Likewise. BMI from the PUS and PHS after did not demonstrate a significant difference (*p* = 0.220, F (1.50) = 0.024, *ηp*^2^ = 0.038 (small effect).

## 6. Discussion

At the time of admission into the BS service, PUS patients had a higher severity profile according to AORC score, frequency of severe obesity, and greater waist circumference, and could be considered initially more at risk when compared to PHS patients.

The study conducted by Guibu et al. [19] showed that the profile of the PUS user population includes more than half having only completed elementary school, and classified as lower middle class; moreover, they present a higher prevalence of chronic diseases, such as hypertension, obesity and diabetes mellitus. Numerous studies have shown [20,21] a relationship between low socioeconomic/educational conditions and an increased incidence of obesity. The phenomenon linking poverty to obesity has been studied intensely, and some data have been appearing in the literature in an attempt to explain this link [22]. Both access to ultra-processed foods and consumption of these foods, with their high caloric densities and low cost, are available to the entire population, as well as sedentary lifestyles, not just among the poor [23]. However, Lewis, Edwards-Hampton, and Ard [24] described in their study that the ability to seek or follow behavioral recommendations related to lifestyle changes, such as diet and physical activity, may be limited in low socioeconomic/educational conditions.

The time elapsed between admission and the pre-surgery evaluation was longer among the PUS patients than that in the PHS patients. This shows that there are prolonged waiting times in PUS, and this finding corroborates the work of Rasera et al. [4], performed at a Brazilian obesity surgery center, showing that the mean interval between an indication for surgery and the procedure is approximately two years. In fact, the literature also shows a disparity between the pre-surgery follow-up of BS in Brazilian health systems, being 14 months on the PUS and two months on the PHS [8].

Due to the long waiting times, patients undergoing BS at PUS may see their pre-surgery profile worsen with time, thereby having more comorbidities and higher BMI when compared with PHS patients. Indeed, in the current study, in the period between admission and pre-surgery evaluation, there was no clinical change in comorbidities associated with obesity in the PUS group, in contrast to a reduction in SAH in the PHS group. Previous studies of Marcon et al. [25], Borisenko et al. [26], and Gushiken et al. [27] also reported that pre-surgery waiting time might lead to weight gain and increased morbidity and mortality. In the study conducted by Marcon et al. [25], the mean waiting time was 24 months, and after enrollment of the patients in the study, five individuals awaiting surgery died.

Collectively, these results reflect the various impairments of the public health system (e.g., human resources, equipment) and derive from a small capacity to supply the demand [9,22].

The reduction of factors associated with cardiometabolic risk in the two groups, in the three-month post-surgery period, may be related to caloric restriction, the altered performance of gastrointestinal hormones, and an improvement in anthropometric measures. In fact, caloric restriction may lead to rapid weight loss due to the immediate decrease in food consumption after BS (to 200–300 kcal/day), as a result of decreased capacity and reduction of stomach emptying rate, difficulty in tolerating intake of foods with high osmolarity, malabsorption of fat, and increased satiety [28]. Mulla et al. [29] described in their study the role of intestinal hormones such as GLP-1, PYY, ghrelin, and oxyntomodulin. Such hormones have anorexigenic actions, promoting early satiety, thus contributing to weight loss. However, it is not clear to what extent these hormones regulate appetite and food intake in the post-surgery period after BS.

Regarding the anthropometric measures, the present study obtained similar results to that of Andersson et al. [30], Szczuko et al. [31], Schiavon [32], and Nassour et al. [33]. The latter states that the reduction of the anthropometric measures is more effective in the first year, mainly in regards to the weight loss. Hemmingsson et al. [34] and Dixon et al. [35] reported that anthropometric measures of BMI and waist circumference provide the necessary results to measure CRF, recommending the use of these anthropometrics as clinical tools to provide information on CRF and inflammatory markers. Andersson et al. [30] concluded that waist circumference could be used to evaluate the reversal of insulin resistance in obese patients after BS-induced weight loss, and suggested that waist circumference could be a valuable parameter to indirectly assess the risk of developing diabetes mellitus or to relapse after surgically-induced weight reduction.

Regarding physical activity, PHS patients in the present study had a significant increase in physical activity level post-surgery and, consequently, were able to increase weight loss. It is also documented in the literature [36,37] that increased activity level after surgery optimizes weight loss compared with sedentary behavior. This finding concludes that there is an association between exercise and success on the treatment of obesity, while a sedentary lifestyle is strongly associated with insufficient post-surgery weight loss. The results of Pereira et al. [38], Elbelt et al. [39], and Jacob et al. [40] show an increase in the interest in exercise during the first post-surgery year with an association with greater weight loss. These findings match that in the present study, where the level of physical activity was higher post-surgery when compared with pre-surgery.

Some limitations of this study should be mentioned, among them: sample selection bias, because data were collected retrospectively from records of patients who underwent BS; a follow-up time of only one year; the lack of additional data that could complement the work, such as the types of surgical techniques that were used; blood pressure values; and additional biochemical tests such as glycosylated hemoglobin and insulin dosage in diabetics.

## 7. Conclusions

On admission to bariatric surgery, public health system patients had a higher severity profile according to their AORC scores and a higher frequency of severe obesity when compared to those treated by the private health system. The time elapsed between admission and the pre-surgery evaluation was longer in the public system when compared with the private system. In this period, there were no clinical changes in the comorbidities associated with obesity in the public system patients in contrast with the private system patients that had a reduction in SAH during this pre-surgery period. At 3, 6, and 12 months after the surgery, there were no differences in the changes of the AORC score between both institutions. In the physical activity levels, an improvement was observed at 12-month post-surgery in both health systems when compared with admission time.

## Figures and Tables

**Figure 1 ijerph-16-04751-f001:**
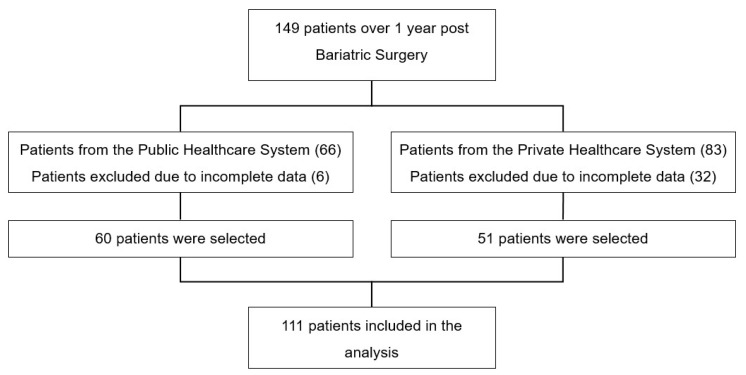
Study Design.

**Figure 2 ijerph-16-04751-f002:**
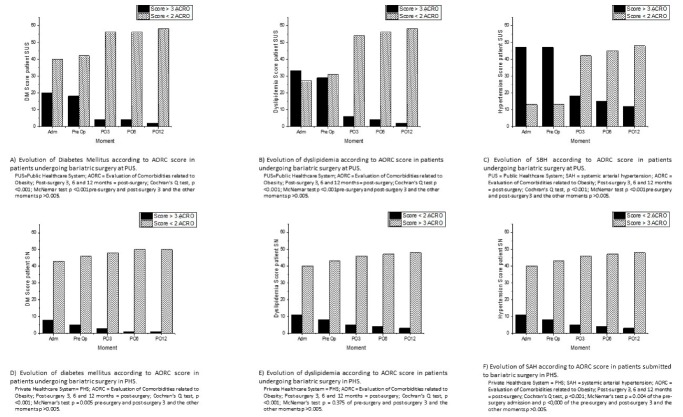
Evolution of according to AORC score in (**A**) Diabetes mellitus PUS; (**B**) dyslipidemia PUS; (**C**) of SAH PUS; (**D**) Diabetes mellitus PHS; (**E**) dyslipidemia PHS; (**F**) SAH in PHS.

**Figure 3 ijerph-16-04751-f003:**
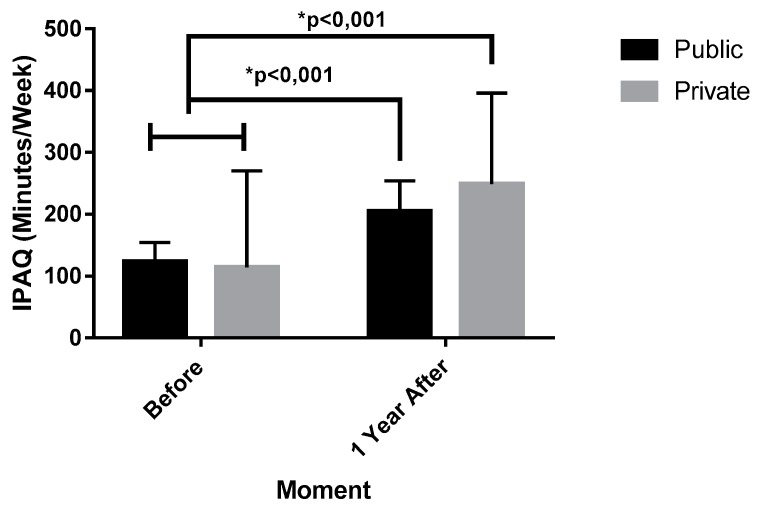
Evolution of physical activity level before and after 1 year of bariatric surgery in PUS and PHS individuals. IPAQ = International Physical Activity Questionnaire; Public = public healthcare system; Private = private healthcare system.

**Figure 4 ijerph-16-04751-f004:**
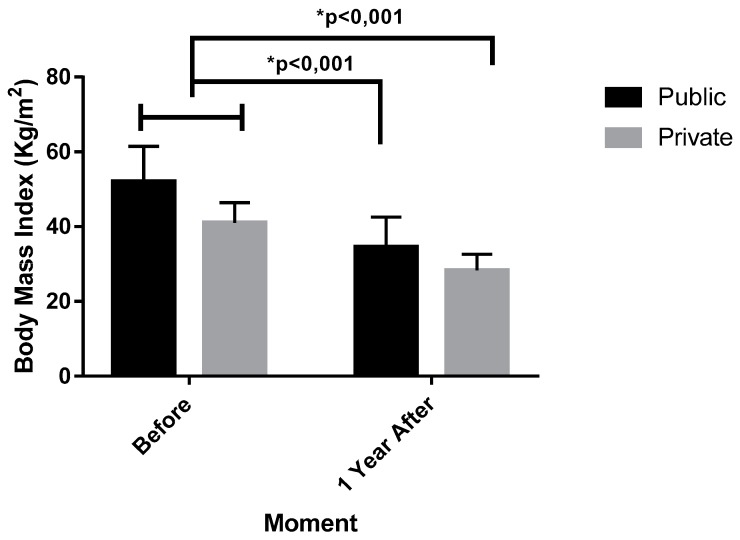
Evolution of Body Mass Index (BMI) before and after 1 year of bariatric surgery in the PUS and PHS individuals.

**Table 1 ijerph-16-04751-t001:** Evaluation of obesity-related comorbidities (AORC).

AORC Score	Diabetes Mellitus
0	Absence
1	Glucose intolerance (≥100 e < 126 mg/DL)
2	Diabetes mellitus (diagnosed)
3	Controlled with oral antidiabetic
4	Insulin therapy
5	Clinical complications
**Dyslipidemias**	
0	Absence
1	Borderline (200–239 mg/DL)
2	Conventional control (diet + physical activity)
3	Single medicinal product
4	Multiple medications
5	Uncontrolled
**Systemic Blood Hypertension**	
0	Absence
1	Borderline values (systolic: 130–139 mmHg, diastolic: 85–89 mmHg)
2	Conventional control (diet + physical activity)
3	Single medicinal product
4	Multiple medications
5	Uncontrolled

**Table 2 ijerph-16-04751-t002:** Characterization of patients undergoing bariatric surgery at PUS and PHS.

Variables	Total *n* = 111	PUS *n* = 60	PHS *n* = 51	*p*
Age (years)	39.6 ± 10.8	42.5 ± 9.4	37.4 ± 12.0	0.048 ^1^
Comorbidities				
SAH	81 (73.0)	50 (83.3)	31 (60.8)	0.008 ^2^
Dyslipidemia	65 (58.6)	39 (65.0)	26 (51.0)	0.135 ^2^
Diabetes Mellitus	39 (35.1)	27 (45.0)	12 (23.5)	0.018 ^2^
**Admission time to the pre-surgery period (months) ^2^**	10 (2; 20)	17 (13; 28.5)	2 (2; 6)	<0.001 ^2^
**Gender**	***n* (%)**	***n* (%)**	***n* (%)**	
Female	80 (72.1)	45 (75.0)	35 (68.6)	0.46 ^3^
**Nutritional status**				<0.001 ^3^
Overweight	1 (0.9)	0	1 (1.9)	
Obesity I	27 (24.3)	2 (3.3)	25 (49.0)	
Obesity II	48 (43.2)	27 (45.0)	21 (41.0)	
Severe obesity III	35 (31.5)	31 (51.6)	4 (7.8)	

The variables were expressed as age in years as mean and standard deviation and for admission time with median and maximum and minimum values. PUS = public healthcare system; PHS = private healthcare system; BMI = body mass index; ACRO = Evaluation of Comorbidities Related to Obesity; SAH = systemic arterial hypertension. ^1^ Student’s t-test; ^2^ Mann–Whitney; ^3^ Fisher exact test.

**Table 3 ijerph-16-04751-t003:** Comparison of the evolution of factors associated with cardiometabolic risk according to the AORC score in patients undergoing bariatric surgery in the PUS and PHS at different moments of care.

AORC SCORE	Diabetes Mellitus			Dyslipidemia			SAH		
	PUS *N* = 60	PHS *N* = 51	*p*	PUS *N* = 60	PHS *N* = 51	*p*	PUS *N* = 60	PHS *N* = 51	*p*
Admission	***n* (%)**	***n* (%)**		***n* (%)**	***n* (%)**		***n* (%)**	***n* (%)**	
≤2	40 (66.7)	43 (84.3)	0.033 ^1^	27 (45.0)	40 (78.4)	<0.0001 ^1^	13 (21.7)	22 (43.1)	0.015 ^1^
≥3	20 (33.3)	8 (15.7)		33 (55.0)	11 (21.6)		47 (78.3)	29 (56.2)	
Pre-surgery									
≤2	42 (70.0)	46 (90.2)	0.009 ^1^	31 (51.7)	43 (84.3)	<0.000	13 (15.4)	31 (3.7)	<0.001 ^2^
≥3	18 (30.0)	5 (9.0)		29 (48.3)	8 (15.7)		47 (78.3)	20 (39.2)	
Post-surgery 3 months									
≤2	56 (93.3)	48 (94.1)		54 (90.0)	46 (90.2)	0.937 ^1^	42 (70.0)	44 (86.3)	0.067 ^2^
≥3	4 (6.7)	3 (5.9)	1	6 (10.0)	5 (9.8)		18 (30.0)	17 (13.7)	
Post-surgery 6 months									
≤2	56 (93.3)	50 (98.0)	0.372 ^2^	56 (93.3)	47 (92.2)	0.811	45 (75.0)	46 (90.2)	0.157 ^1^
≥3	4 (6.7)	1 (2.0)		4 (6.7)	4 (7.8)		15 (25.0)	5 (9.8)	
Post-surgery 12 months									
≤2	58 (96.7)	50 (98.0)	1	27 (45.0)	40 (78.4)	1 ^1^	48 (80.0)	47 (92.2)	
≥3	2 (3.3)	1 (2.0)		33 (11.0)	11 (21.6)		12 (20.0)	4 (7.8)	0.069 ^1^

Values expressed as *n* (%); ACRO = Evaluation of Comorbidities Related to Obesity; values expressed as *n* (%); *p* = significance level; AORC = Evaluation of Comorbidities Related to Obesity; ≤2 = absence of comorbidities; ≥3 = presence of comorbidities (diabetes, dyslipidemia and SAH). ^1^ Chi-square test; ^2^ Fisher’s exact test.

**Table 4 ijerph-16-04751-t004:** Evolution and anthropometric comparison of individuals in the sample submitted to bariatric surgery in the PUS and PHS.

Variables	PUS *N* = 60	PHS *N* = 51	*p*
**Pre-surgery Nutritional Status ^1^**	***n* (%)**	***n* (%)**	<0.0001
Overweight	0	1 (1.9)	
Obesity I	13 (21.6)	34 (66.6)	
Obesity II	20 (33.3)	14 (27.5)	
Severe Obesity III	27 (44.9)	2 (3.8)	
**Post-surgery Nutritional Status 12 months ^1^**			<0.0001
Eutrophic	3 (5.0)	9 (17.6)	
Overweight	9 (31.7)	29 (56.9)	
Obesity I	9 (15.0)	10 (19.6)	
Obesity II	14 (23.3)	2 (3.9)	
Severe Obesity III	15 (24.9)	1 (1.9)	
	**X ± DP**	**X ± DP**	
Weight loss percentage	63.3 ± 25.6	80.0 ± 25.6	0.002
Weight admission (kg)	133.8 ± 28.1	114.7 ± 22.1	<0.0001
Pre-surgery weight (kg)	126.4 ± 27.0	109.5 ± 19.5	<0.0001
Current weight (kg)	90.3 ± 22.8	79.3 ± 16.6	<0.0001
BMI admission (Kg/m^2^)	51.5 ± 9.4	41.1 ± 5.4	<0.0001
BMI pre-surgery (Kg/m^2^)	48.7 ± 9.2	39.2 ± 4.4	<0.0001
Post-surgery BMI (Kg/m^2^)	34.8 ± 8.7	28.4 ± 4.3	<0.0001
Waist circumference (cm)	128.17 ± 15.3	114.3 ± 13.0	0.033
Final waist circumference (cm)	105.7 ± 17.6	99.2 ± 12.0	0.028
Waist/initial height ratio	0.78 ± 0.09	0.68 ± 0.66	<0.0001
Waist/final height ratio	0.64 ± 0.13	0.59 ± 0.06	0.010

The variables were expressed in BMI = body mass index (kg/m^2^). pre-surgery, and post-surgery 12 months; weight in kg; X ± SD = mean and standard deviation. ^1^ Fisher’s exact test and the others were Student’s t-test applied with significance level *p* < 0.005.

**Table 5 ijerph-16-04751-t005:** Average referring to the biochemical parameters of patients undergoing bariatric surgery in PUS and PHS.

PUS				PHS		
Variables	Preoperative	Postoperative 3	Postoperative 6	Preoperative	Postoperative 3	Postoperative 6
Fasting Blood Glucose	90.4 ± 15.4	87.3 ± 13.5	79.7 ± 8.6	96.8 ± 23.3	85.3 ± 8.7	83.3 ± 5.8
Triglycerides	134.2 ± 76.7	110.6 ± 47.8	81.4 ± 30.7	155.9 ± 72. 3	112.6 ± 48.6	86.8 ± 24.6
Total cholesterol	182.5 ± 37.5	174.6 ± 35.3	148.3 ± 27.9	203.0 ± 40.2	175.4 ± 34.5	174.0 ± 30.3
LDL	111.7 ± 29.6	111.2 ± 29.0	88.7 ± 28.2	127.3 ± 32.8	113.3 ± 25.8	101.4 ± 29.1
HDL	45.2 ± 9.4	44.2 ± 10.5	45.0 ± 11.4	41.5 ± 9.0	40.7 ± 9.9	48.7 ± 13.6

Public healthcare system (PUS) and private healthcare system (PHS); LDL = low density lipoprotein; HDL = high density lipoprotein.

**Table 6 ijerph-16-04751-t006:** Time variation (pre and postoperative 6 and 12 months) compared with PUS and PHR users for biochemical parameters.

Variables	*p* *	Size of Effect (*η*^2^)	Observed Power
Time Effect for:			
HDL	0.008	0.090	0.807
LDL	<0.0001	0.340	1.000
total cholesterol	<0.0001	0.284	1.000
Triglycerides	<0.0001	0.455	1.000
Fasting glucose	<0.0001	0.197	1.000
Effect between groups to:			
HDL	0.014	0.080	0.750
LDL	0.046	0.058	0.595
Total cholesterol	<0.001	0.063	0.920
Triglycerides	0.363	0.020	0.225
Fasting glucose	0.057	0.029	0.563

Public healthcare system (PUS) and private healthcare system (PHS); LDL = low density lipoprotein; HDL = high density lipoprotein; * ANOVA with repeated measurements over time.

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
