# Peer review of "A Retrospective Study about the Differences in Cardiometabolic Risk Indicators and Level of Physical Activity in Bariatric Surgery Patients from Private vs. Public Units"

_ijerph, 2019, doi:10.3390/ijerph16234751_

Round 1
Reviewer 1 Report
The paper entitled “A 12-month follow-up on anthropometrics, physical activity and cardiometabolic risk of bariatric surgery patients in private vs. public units” is properly planned and well-designed study evaluating the changes in anthropometrics, cardiometabolic and physical activity patterns of bariatric surgery patients on PUS and on PHS, pre- and up to 12-month post-surgery. The authors of the study concluded that: 3, 6, and 12 months after the surgery, there were no differences in the changes of the evaluation of obesity-related comorbidities score between Public Healthcare System and Private Healthcare System; according to physical activity, an improvement was observed at 12-month post-surgery in both health systems, when compared with admission time. Overall, this manuscript is very interesting and contains novel aspects. However, there are some parts of the manuscript that should be improved: Figure no. 1 is prepared extremely carelessly; The manuscript of the work should be supplemented with patient characteristics based on routine laboratory test results (e.g. glycemia, lipid profile/panel, blood smear, results of liver enzymes activity i.e. ALT, AST); Applied in this study BMI has serious limitations. i.e. difficulty in distinguishing between lean and fat mass; Difficulties in identifying body fat distribution; Additionally, variability of validity across gender, races and ethnicities, and age groups. Therefore, in order to assess the results of research more efficiently, other anthropometric indexes, such as WHR (waist to hip ratio) WHtR (waist-to-height ratio).
Author Response
Overall, this manuscript is very interesting and contains novel aspects. However, there are some parts of the manuscript that should be improved:
Figure no. 1 is prepared extremely carelessly;Answer: Figure 1 has been changed for a better understanding.
The manuscript of the work should be supplemented with patient characteristics based on routine laboratory test results (e.g. glycemia, lipid profile/panel, blood smear, results of liver enzymes activity i.e. ALT, AST);
Answer: Patient characteristics were entered based on routine laboratory test results. Only liver-related variables were not entered because they were not analyzed.
Applied in this study BMI has serious limitations. i.e. difficulty in distinguishing between lean and fat mass; Difficulties in identifying body fat distribution; Additionally, variability of validity across gender, races and ethnicities, and age groups. Therefore, in order to assess the results of research more efficiently, other anthropometric indexes, such as WHR (waist to hip ratio) WHtR (waist-to-height ratio).
Answer: It has been changed for a better understanding.
In the attachment you can check the new version of the article after considerations.
Thank you for your contributions, looking foward to hear from you.

Reviewer 2 Report
The study is very interesting and well designed and discussed.
I only have one minor suggestion that may improve the quality of the article.
- In the introduction, the authors dont't talk about the importance of several anthropometric measures of abdominal visceral obesity and their association with coronary atherosclerosis; please see these references (PMID: 31301983, PMID: 29680968) and comment these in the Introduction.
Author Response
In the introduction, the authors don’t talk about the importance of several anthropometric measures of abdominal visceral obesity and their association with coronary atherosclerosis; please see these references (PMID: 31301983, PMID: 29680968) and comment these in the Introduction.ANSWER: A paragraph has been inserted in the introduction for a better understanding.
In the attachment you can check the new version of the article after considerations.
Thank you for your contributions, looking foward to hear from you.

Reviewer 3 Report
The authors present a study to determined wether there are differences in the treatment outcome after bariatric surgery in the public and private health care systems in brazil. The study represents a comprehensive statistical analysis of questionnaire data (IPAQ) of a total of 111 patients. The authors conclude that the difference in waiting times leading up to bariatric surgery is the major disadvantage in the public system compared to the private system whch also leads to worse severity profles at the time of surgery in the public system. This is slightly in contrast to the title stating a 12-month follow-up, hence, the time post-surgery. The authors also conclude that there were no significant differences in the 12-months post-surgery. Therefore, the title should be clarified since the time pre-surgery is in fact presented as the most significant factor.
While the statistical analysis itself appears to be sound the >10 authors seem to have failed to proof read the manuscript before submission which doesn't fit to the otherwise high quality of the study.
Formal problems
The manuscript bares multiple formal errors up to the point that it is not possible to review the content of the manuscript and it is surprising to me that the authors submit such a desolate document to peer-review.
font-size changes in the abstract already Figure1 is messed up spaces missing in Table1 Figures 8 and 9 are called "Figura" and are just copy and pasted. They show the exact same barplots on the exact same scale. However, the y-axis label was changed. On pages 6 and 7 some of the text is duplicatedWhile I think the study is eligible for publication given the extensive formal weakness I have to recommend reconsideration after major revisions.
Author Response
All considerations have been met. The article went through an extensive review of english by Enago® and we believe the article has better reading potential now.
In the attachment you can check the new version of the article after considerations.
Your considerations enhanced the article, thank you.
Looking forward to hear from you.

Round 2
Reviewer 3 Report
The authors made significant changes that already greatly helped to increase the overall quality and clarity of the manuscript. However, not all improvements managed to fully address the problems and I am irritated by the fact that these remaining problems were apparently missed by the authors, who need to step up their quality measures for paper submissions!
1. Figure 1 is still messy.
2. Spaces are still missing in Table 1 (see 3).
3. Figure 2 reaches over multiple pages. I'd recommend to split it up in more manageable parts.
4. In Figure 4, p-value indicators seem to be copied from Figure 3 and don't fit bar heights in Figure 4. Not dramatic but could be improved.
Author Response
All changes were made as requested.
Thank you again, looking forward to hear from you.
